# Enhancing the Catalytic Activity of Type II L-Asparaginase from *Bacillus licheniformis* through Semi-Rational Design

**DOI:** 10.3390/ijms23179663

**Published:** 2022-08-26

**Authors:** Yawen Zhou, Linshu Jiao, Juan Shen, Huibing Chi, Zhaoxin Lu, Huawei Liu, Fengxia Lu, Ping Zhu

**Affiliations:** 1College of Food Science and Technology, Nanjing Agricultural University, Nanjing 210095, China; 2Institute of Food Safety and Nutrition, Key Lab of Food Quality and Safety of Jiangsu Province-State Key Laboratory Breeding Base, Jiangsu Academy of Agricultural Sciences, Nanjing 210014, China

**Keywords:** L-asparaginase, protein engineering, saturation mutagenesis, catalytic efficiency, molecular dynamics simulation

## Abstract

Low catalytic activity is a key factor limiting the widespread application of type II L-asparaginase (ASNase) in the food and pharmaceutical industries. In this study, smart libraries were constructed by semi-rational design to improve the catalytic activity of type II ASNase from *Bacillus licheniformis*. Mutants with greatly enhanced catalytic efficiency were screened by saturation mutations and combinatorial mutations. A quintuple mutant ILRAC was ultimately obtained with specific activity of 841.62 IU/mg and *k*_cat_/*K*_m_ of 537.15 min^−1^·mM^−1^, which were 4.24-fold and 6.32-fold more than those of wild-type ASNase. The highest specific activity and *k*_cat_/*K*_m_ were firstly reported in type II ASNase from *Bacillus licheniformis*. Additionally, enhanced pH stability and superior thermostability were both achieved in mutant ILRAC. Meanwhile, structural alignment and molecular dynamic simulation demonstrated that high structure stability and strong substrate binding were beneficial for the improved thermal stability and enzymatic activity of mutant ILRAC. This is the first time that enzymatic activity of type II ASNase from *Bacillus licheniformis* has been enhanced by the semi-rational approach, and results provide new insights into enzymatic modification of L-asparaginase for industrial applications.

## 1. Introduction

L-asparaginase (ASNase, E.C.3.5.1.1) is a specific hydrolase that catalyzes the deamination of L-asparagine to L-aspartic acid and ammonia [1]. L-asparaginases are classified into I and II subtypes based on their structure and intra- and extracellular location [2]. Both types are widely found in animals, plants, and microorganisms [3,4,5]. Among them, type II ASNase has attracted particular attention as an anticancer medication in treating acute lymphoblastic leukemia (ALL) and lymphosarcoma [3] for its excellent substrate affinity. In addition, type II ASNase can effectively reduce the level of potentially carcinogenic acrylamide without affecting the appearance, quality, and taste of the product in food processed using a high-temperature [6,7]. Unfortunately, the applications of type II ASNase are often constrained by low catalytic efficiency, low substrate specificity, and low stability [8]. Thus, improving enzyme performance for type II ASNase has great practical significance for its clinical and food industrial applications.

Structure-based rational design is a powerful protein engineering approach for improving enzymatic properties of ASNases [9]. To achieve precise design for enzymatic improvement, sufficient knowledge of protein structure has to be firstly gained as the basic support [10]. Recently, crystal structure of a human L-asparaginase-like protein has been resolved and used as the foundation for rational design of catalytic activity improvement with a 6-fold increase by residue mutation of the catalytic cavity [11]. In addition, homology modeling of *Bacillus subtilis* ASNase has been applied for directing site-directed mutagenesis of residues near the catalytic cavity with higher specific activity and hydrophilicity of surface charge redistribution [12]. However, due to enzymes with intrinsically dynamic distal residues that also play key roles in promoting catalytic activity [13], mutated regions adjacent to the catalytic cavity and subunit interfaces can cause the limitation of site-directed mutagenesis. Previous research has indicated that mutations of residues far from the active sites affect the catalytic efficiency of prolyl-tRNA synthase [14]. These distal residues contribute considerably to the increased catalytic efficiency by maintaining the intrinsic flexibility of protein. Another study has reported that mutating the distal position assesses ornithine transcarbamoylase with enhanced enzymatic efficiency [15]. Thus, distal residues may play an important role in enzymatic improvement of type II ASNase.

Computational tools, which are incredibly beneficial for improving screening efficiency, have expanded the selection scope of mutation sites [10,16]. HotSpot Wizard server can be used for mutation site identification and library design [17], which has predicted hotspots of mutations for increasing enzyme activity [18,19,20]. EVcoupling web server can provide evolutionary constraints (EC) to infer correlations between amino acids at different sequence positions [21] and affect enzyme structure and function [22], which has been proved to be an effective approach to improving catalytic activity [23,24]. However, enzymatic improvement of type II ASNase by rational design combing structural information and bioinformatics tools has been rarely reported. Considering the significance of enzyme performance for type II ASNase application, we attempted to improve catalytic activity of *Bacillus licheniformis* ASNase (BLASNase), which is a novel type II ASNase from GRAS source with good thermal stability, but low catalytic efficiency hinders its clinical and food industrial applications.

Herein, we present an effective approach for improving catalytic activity of BLASNase. This approach proceeds by screening active center residues through structural analysis and sequence comparison, and identifying distal residues through bioinformatics servers (HotSpot Wizard and EVcoupling), which allows the construction of smart libraries for directly performing mutations at specific sites with reduced screening workload. Further, computer-aided designs such as homology modeling, molecular docking, and molecular dynamics simulation were combined to reveal the relationships of protein conformational change and increased enzyme activity by modified residues.

## 2. Results and Discussion

### 2.1. Generation of Mutation Sites

COACH, Hotspot Wizard 3.0, and EVcoupling are general servers for predicting possible functional sites. Sequence conservation of BLASNase was evaluated through the ConSurf server (Figure 1A). The amino acid sequence of BLASNase was compared with that of ASNases from three other sources with relatively higher enzymatic activity (Figure 1B). After comparison, the active site of ASNases from different microorganisms displayed highly conservative, composed of Thr62, Tyr76, Ser109, Thr142, Asp143, and Lys215. Since sites around the active center may affect enzymatic catalytic activity [25], nonconserved residues within 5 Å around the active site within the catalytic pocket were screened via the COACH server and combined with multiple sequence alignment (MSA) results. As a result, L203, K218, T223, and T325 were selected for saturation mutation. Meanwhile, Hotspot Wizard 3.0 web server, which provides a list of functional hot spots for protein analysis, was employed to select Q110 and A180 for saturation mutation by combining the mutability and conservation of hot spots. Finally, after entering the amino acid sequence into the EVcoupling server for evolutionary coupling analysis, two pairs of coupling sites, E102/R123 and V104/R57, were selected for cosaturation mutation for analyzing the coupling strength between residues and the position in the spatial structure. The constructed saturation and combinatory mutants were ultimately generated for further analysis by these approaches.

### 2.2. Screening of Mutant Libraries

To evaluate the catalytic activity, all generated mutants were highly purified for screening validations. As a result, five single-point mutants (E102I, V104L, A180R, T223A, and T325C) displayed relatively high enzyme activities. Since combinatorial mutagenesis can assess possible additive or even synergistic effects [26], double, triple, quadruple, and quintuple mutants were further constructed based on the results of single mutants. Surprisingly, a quintuple mutant, ILRAC (E102I/V104L/A180R/T223A/T325C), was obtained with remarkably enhanced enzymatic activity after screening. As shown in Figure 2A, the purity and molecular weight (about 46 kDa) of ILRAC were quite similar to its single-point mutants and wild-type. However, the specific activity of mutant ILRAC reached 841.62 IU/mg (Figure 2B and Appendix A), which was 4.24-fold that of wild-type (198.40 IU/mg), indicating a significant improvement in enzymatic activity. Additionally, the enzyme activity of our ILRAC mutant was even higher than that of the best mutant identified by Sudhir and Long [12,27]. These results preliminarily confirmed that semi-rational design combing structural information and bioinformatics tools was effectively applied for enhancing enzymatic activity of BLASNase. It could also provide a basis for molecular modification of L-asparaginase from other sources.

### 2.3. Enzymatic Characterization of Wild-Type and Mutant ILRAC

To look into the detailed enzymatic properties of mutant ILRAC, the optimum reaction pH and temperature of mutant ILRAC and wild-type BLASNase were firstly determined by analyzing their relative activity under different pH gradients and temperatures of pH 3.0–13.0 and 30–60 °C. The optimal pH of wild-type was 8.0 while the optimal pH of ILRAC slightly shifted to 9.0 (Figure 3A). Of note, the ILRAC had better pH stability, which kept over 80% initial activity after 24 h incubation at pH 6.0–11.0 (Figure 3B). On the contrary, wild-type could retain less than 80% initial activity at the same pH conditions. As wide pH range and excellent pH stability are required for high pH baked goods [28], our mutant ILRAC with pH stability could be a potential agent in industrial food application The optimum reaction temperatures of wild-type and mutant ILRAC were both 40 °C (Figure 3C). As expected, mutant ILRAC displayed the same high enzyme activity as the thermally stable wild-type, in which both maintained more than 80% initial activity for the range 30 to 60 °C. Interestingly, mutant ILRAC kept over 50% initial activity during 24 h incubation at 60 °C and was slightly better than wild-type (Figure 3D). The temperature stabilities of both wild-type and mutant ILRAC were even much better than those of clinical ASNase from and *E. chrysamthemi* (inactivated after heat treatment at 60 °C for 7.5 min) and *E. coli* (exhibited 40% initial activity after pre-incubation at 60 °C for 10 min) [29,30]. Generally, a high thermostability and long half-life are beneficial for food and pharmaceutical applications of L-asparaginase [31]. Therefore, enhanced thermostability would be helpful for the application of mutant ILRAC.

In general, low *K*_m_ and high *k*_cat_ are critical factors for ASNase therapeutic products [31]. The enzyme kinetic parameters of wild-type and mutant ILRAC were further estimated. As shown in Table 1, the *K*_m_ value of mutant ILRAC decreased to 1.45 mM from 2.33 mM of wild-type, and the *k*_cat_ value of mutant ILRAC increased to 778.87 min^−^^1^ from 197.95 min^−^^1^ of wild-type. Consequently, the *k*_cat_/*K*_m_ of mutant ILRAC increased 6.32-fold more than that of wild-type. The *k*_cat_/*K*_m_ value of the mutant ILRAC was higher than some other ASNases from different source, such as *Bacillus megaterium* H-1 (3.39 min^−1^-mM^−1^) and *Bacillus subtilis* (397.8 min^−1^-mM^−1^) [32,33]. Although the recorded *K*_m_ value of mutant ILRAC was still higher than clinical ASNase from *E. coli* (10 μM) and *E. chrysanthemi* (12 μM), it was much lower than other types II ASNases from multiple sources such as *B. subtilis* (5.29 mM), *Bacillus licheniformis* (49.995 mM), *Pseudomonas aeruginosa* (10.904 mM), and *Streptomyces fradiae* (10.07 mM) [34,35,36,37]. Altogether, our rational design strategy appears to have improved the catalytic efficiency and substrate affinity of mutant ILRAC.

### 2.4. Structure Stability Analysis of Wild-Type and Mutant ILRAC

In the aforementioned study, mutant ILRAC has been proved to gain enhanced stability by the rational design approach. Therefore, we hypothesized that the enhanced stability of mutant ILRAC might be caused by structural differences from wild-type ASNase. As circular dichroism determines the alterations to the secondary structure of proteins, circular dichroism spectroscopy analysis was performed for mutant wild-type and ILRAC. The percentages of α-helice of mutant ILRAC increased to 34.5% from 26.0% of wild-type, and the random coils decreased to 31.4% from 39.6% of wild-type (Table 2). Since α-helice improves the rigidity of the protein for maintaining the stability of protein structure [38], and random coils increases the flexibility of the protein for facilitating protein stabilization [39], the increased α-helice and decreased random coils were obtained in mutant ILRAC. These results initially confirmed that increased structural rigidity of mutant ILRAC could contribute to enhanced thermal stability compared to wild-type ASNase.

To further look into the structural stability, molecular dynamic simulations were introduced for mutant ILRAC and wild-type ASNase. As the determination of the root mean square deviation (RMSD) ensures the equilibrium of the system during the simulation [40], both structures of mutant ILRAC and wild-type reached equilibrium after 100 ns simulation (Figure 4), demonstrating that the results of both enzymes could be used for subsequent analysis. Generally, delayed equilibrium indicates lower stability [41]; mutant ILRAC displayed advanced equilibrium for higher stability. In addition to RMSD values, the number of hydrogen bonds, radius of gyration (Rg), and solvent-accessible surface area (SASA) can also reflect the protein stability [42]. The number of intramolecular hydrogen bonds is positively correlated with protein stability. The average intramolecular hydrogen bond numbers in mutant ILRAC and wild-type systems were 812.60 and 810.63, respectively (Table 3), indicating enhamced stability of mutant ILRAC. These computational results were consistent with our experimental observations. The Rg exhibits the compactness of the protein structure; a larger Rg represents a looser protein structure [43]. The average Rg values of mutant ILRAC and wild-type were 20.35 and 20.38 Å, respectively, suggesting similar Rg values of mutant ILRAC and wild-type for their both dense and stable complexes. Higher SASA values normally indicate increased stretching of the protein [44]. The average SASA values were 36,675.61 and 36,276.87 Å^2^ for mutant ILRAC and wild-type, respectively, indicating ILRAC had increased the SASA value for more expanded structure for keeping thermal stability.

### 2.5. Substrate Affinity Analysis of Wild-Type and Mutant ILRAC

To assess the affinity between BLASNase and its substrate, the binding free energy was calculated by MD trajectory for mutant ILRAC and wild-type. As shown in Table 4, the energy decomposition showed that electrostatic energy was the primary contributing energy, followed by van der Waals energy and nonpolar solvation energy. When compared to wild-type, the binding free energy of mutant ILRAC with L-asparagine was lower, implying that mutant ILRAC had stronger substrate binding affinity than wild-type. This result provided the support for our previous experimental data showing that ILRAC had 6-fold increased *k*_cat_/*K*_m_ compared to wild-type.

The free energy contributions are decomposed to assess the contributions of individual residues to substrate binding. Figure 5 shows the top ten residues contributing to enzyme and substrate binding capacity in the wild-type and mutant ILRAC. It is worth noting that the strongest binding contributions of Ser109 and Thr62 in mutant ILRAC had larger values than those of wild-type. In addition, Ser300 in mutant ILRAC displayed higher intensity than in wild-type. Of note, Asp143 played a significant role in substrate binding in mutant ILRAC, which was not found in wild-type. As Asp143 is situated at the interface of the two subunits and acts as an anchor for the substrate [45,46]; therefore, mutant ILRAC could have enhanced interaction with substrate, which contributed the increased enzyme activity.

Furthermore, we sampled the binding conformation of wild-type and mutant ILRAC separately with L-asparagine at 100 ns after simulations and analyzed their binding patterns. As shown in Figure 6A, substrate and enzymes were both well stacked. The mutated sites of ILRAC did not affect the relative positions of enzyme or substrate but slightly differed in detail. Interestingly, mutant ILRAC had a loop of Chain C (295-GSGNSVS-302) with an obvious movement to the substrate binding site compared with wild-type. This loop movement led to new hydrogen bond formations between Asn298 and Ser300 from the loop of Chain C with the substrate (Figure 6C,D). The mutated sites strengthened the binding interaction with mutant ILRAC to substrate. Since substrate binding interaction is related to catalytic effect [47], the firm substrate binding could lead to the increased catalytic activity of mutant ILRAC.

Continuingly, the substrate binding pockets were determined for wild-type and mutant ILRAC. Figure 7A,B and Table 5 show the variations of their substrate binding pockets. After the mutation, the volume of substrate binding pocket increased from 1216.88 to 1248.40 Åm^3^, the surface area expanded from 1074.33 to 1100.43 Åm^2^, and the depth reduced from 28.61 to 28.18 Åm. The larger pocket facilitates substrate binding to the active site [48]. Thus, mutant ILRAC easily hydrolyzed the substrate for increased catalytic efficiency. Notably, as shown in Figure 7C–F, the T223 and T325 sites of ILRAC were located on the loop around the active site. The small amino acid (Ala and Cys) mutations of Thr weakened the hydrogen bonding and increased the flexibility of the loop near the active site in ILRAC, allowing more substrates to enter substrate pockets and participate in the catalytic reaction for increased catalytic activity. The E102 and V104 sites of ILRAC laid on the β-sheet and the loop adjacent to 102nd (Figure 8A,B). The mutations reduced the hydrogen bonding in this region with disappeared hydrogen bonding of E102 to R123 and E102 to A58, the making β-sheet and loop more flexible. The A180 site of ILRAC was situated on the α-helix (Figure 8C,D). Previous reports have demonstrated that the total helicity of protein affects its stability for catalytic activity [49,50]. Mutated A180R formed an additional hydrogen bond with Y183, making ILRAC a tighter structure for improved catalytic activity. Altogether, the strong substrate binding could contribute the enhanced catalytic activity of mutant ILRAC.

## 3. Materials and Methods

### 3.1. Strains and Reagents

The recombinant plasmid pET-30a(+)-BLASNase harboring type II L-asparaginase gene (ansA) from Bacillus licheniformis and *E. coli* BL21 (DE3) strain were preserved in our laboratory. The gene was PCR-amplified using the 2× Hieff Canace^®^ PCR Master Mix from Yeasen Biotech (Shanghai, China). The DNA cloning kit and ClonExpression^®^ II One Step Cloning kit were obtained from Vazyme (Nanjing, China). DpnI was acquired from Takara Biotechnology (Dalian, China). The Bradford Protein Assay Kit was obtained from Beyotime (Shanghai, China). Primers were synthesized from GenScript (Nanjing, China). Solarbio (Beijing, China) provided isopropyl β-d-thiogalactopyranoside (IPTG) and Kanamycin. L-Asparagine, trichloroacetic acid, and all other reagents were from Aladdin (Shanghai, China). All chemicals were analytical grade.

### 3.2. Bioinformatics Analysis and Selection of Mutation Sites

The BLASNasestructures were modeled with SWISS-MODEL and visualized with Pymol. Furthermore, the 3D structure of L-asparagine (PubChem ID: 6267) was downloaded from the PubChem database. The ligand and enzyme were docked using AutoDockTools v1.5.6 [51]. The COACH server was employed to predict protein–ligand binding sites [52] and the Hotspot Wizard 3.0 was used for automated identification of hotspots whose modification might improve the properties of BLASNase [17]. The EVcoupling server was used to compute evolutionary coupling residue pairs (ECs) from evolutionary sequence covariation. ECs were derived from the maximum entropy model via pseudo-likelihood maximization direct-coupling analysis (PLM) [16]. Multiple sequence alignment (MSA) was generated from the UniProt database by the JackHMMER algorithm. The degree of amino acid conservation in proteins was determined using the Consurf server [53].

### 3.3. Saturation Mutagenesis

The recombinant plasmid pET-30a(+)-BLASNase served as the template for saturation mutagenesis. The mutagenesis primers are shown in Appendix A. The plasmid fragments were amplified by PCR and digested with DpnI to remove the primary template. The purified products were ligated and transformed into BL21 (DE3)-competent cells. The positive mutants were selected according to the procedure used by Lu et al. [32]. The positive mutants screened were sequenced by AENTA (Suzhou, China).

### 3.4. Construction of Combination Mutants

Based on the results of saturation mutagenesis, double-point mutants were constructed using the positive single-point mutant plasmid with the highest enzyme activity as a template. Subsequently, the double-point mutant with the highest enzyme activity was used as a template to construct the triple-point mutant. Similarly, quadruple and quintuple mutants were constructed.

### 3.5. Protein Expression and Purification of BliansA

A single recombinant colony was inoculated into LB medium with 50 μg/mL kanamycin at 37 °C and 180 rpm. Protein expression was induced with 100 mg/mL IPTG at 16 °C for 16 h when OD_600_ reached 0.6–0.8. The cells were centrifuged and resuspended in 50 mM PBS (0.3 M NaCl, pH 8.0). The crude enzyme mixture was obtained from cells after ultrasonication and centrifugation and the target protein was eluted through nickel-affinity chromatography. Note that these purification steps were carried out at 4 °C. The concentration of the enzyme liquation was measured using the Bradford method. Finally, SDS-PAGE was used to assess the purity and molecular weight of recombinant enzymes.

### 3.6. L-Asparaginase Activity Assay

The enzyme activity of L-asparaginase was determined by Nessler’s method [54]. One hundred microliters of suitably diluted enzyme was mixed with 100 μL L-asparagine (189 mM) and 700 µL of 50 mM PBS buffer (pH 8.0). The mixture was incubated at 37 °C for 10 min and then 100 µL of 25% trichloroacetic acid was added to terminate the reaction. Next, 40 µL of the reaction supernatant was added to 100 µL of Nessler’s reagent and 860 µL of deionized water for color development. After 10 min, the absorbance was measured at OD_436_ nm to determine the amount of ammonia released. The amount of enzyme required to release 1 µmol NH_3_ per minute under assay conditions is defined as one international activity unit (IU) of L-asparaginase.

### 3.7. Determination of Kinetic Parameters

The maximal velocity (V_max_), Michaelis−Menten constant (*K*_m_), and catalytic constant (*k*_cat_) values of enzymes were determined in 50 mM PBS buffer (pH 8.0) with various concentrations of L-Asparagine (2 to 20 mM) and 100 µL enzyme at 37 °C for 2 min. Data were calculated by using Lineweaver–Burk plots. All measurements were done in triplicate.

### 3.8. Effect of pH and Temperature on Enzyme Activity and Stability

The optimum reaction temperature was measured within the range 30–60 °C. The thermal stability at 60 °C was measured at pH 8.0 and purified enzymes were placed in a 60 °C water bath. Samples were placed in a 60 °C water bath for 120 min and taken at 20 min intervals, followed by a 10 min ice bath treatment. The enzymatic activity at 0 h was classified as 100%.

The optimum pH was determined in the range 3.0 to 13.0 using the enzyme assay protocol. Enzymes were incubated in various buffers at 4 °C for 24 h to assess their pH stability. The relative activity was determined as the sample activity at 24 h divided by the initial sample activity at 0 h. The buffer formulation for pH 3.0–11.0 followed Yim’s method [55], while 50 mM potassium chloride sodium hydroxide buffer was used for pH 12.0–13.0.

### 3.9. Circular Dichroism Spectroscopy Analysis

Circular dichroism (CD) can determine changes in protein secondary structure between wild-type and mutants. Protein concentration was adjusted to 100 μg/mL and scanned in the wavelength range 190–250 nm. The percentage of secondary structures was quantified by DICHROWEB [56,57].

### 3.10. Molecular Dynamic (MD) Simulations

Next, the potential mechanism underlying increased enzyme activity of the mutant ILRAC needed to be further explored. We performed all-atom MD simulations of protein–ligand complexes using the AMBER 18 package. The ff14SB and GAFF2 force fields were used for proteins and ligands [58,59]. A truncated octahedral TIP3P solvent box was added to the system at a distance of 10 Å [60]. Na^+^ and Cl^−^ were subsequently added to neutralize the system. The two protein–ligand complexes were solventized in an octahedral TIP3P water box, subsequently neutralized by adding Na^+^ and Cl^−^. Next, this system was energy minimized with the steepest descent in 2500 steps and conjugate gradient in 2500 steps. The isochoric−isothermal (NVT) ensemble and isothermal–isovolumetric (NPT) ensemble were performed at 298.15 K for 500 ps, respectively. Finally, trajectories of 100 ns were obtained under the isobaric–isothermal (NPT) ensemble. During the simulations, the system pressure was 1 atm, and the integration step was 2 fs. Traces were stored at 10 ps intervals for further calculations. Protein structures were entered into the Proteinsplus online server to predict the substrate binding pocket and calculate properties [61,62].

The binding free energy between the protein and the ligand was determined by the MM/GBSA approach [63,64]. The MD trajectory 95–100 ns was employed to calculate with the following equation:ΔG_bind_ = ΔG_complex_ − (ΔG_receptor_ + ΔG_ligand_)

                                = ΔE_internal_ + ΔE_VDW_ + ΔE_elec_ + ΔG_GB_ + ΔG_SA_(1)

In Equation (1), ΔGbind represents binding free energy, ΔE_internal_ represents internal energy, ΔE_VDW_ represents van der Waals interaction, and ΔE_elec_ represents electrostatic interaction. The internal energy includes bond energy (E_bond_), angular energy (E_angle_), and torsion energy (E_torsion_). ΔG_GB_ and ΔG_SA_ are collectively referred to as solvation free energy. ΔG_GB_ is the polar solvation free energy and ΔG_SA_ is the nonpolar solvation free energy.

## 4. Conclusions

In this study, we explore a novel semi-rational and low workload approach for catalytic activity improvement of *Bacillus licheniformis* type II ASNase. To the best of our knowledge, this study provides the first investigation of enhancing enzymatic activity of type II ASNase from *Bacillus licheniformis* by the semi-rational design process. The lowest *K*_m_, highest specific activity, and highest thermal stability of type II ASNase from *Bacillus licheniformis* are firstly achieved in our study. Further investigation indicates that high structure stability and strong substrate binding are beneficial for the improved enzymatic performance of mutant ILRAC. Our ILRAC can be used as a potential agent instead of wild-type ASNase for clinical and food industrial applications. It has great potential for application in the pharmaceutical and food sectors. Meanwhile, our results can provide a basis for molecular modification of asparaginases from other sources.

## Figures and Tables

**Figure 1 ijms-23-09663-f001:**
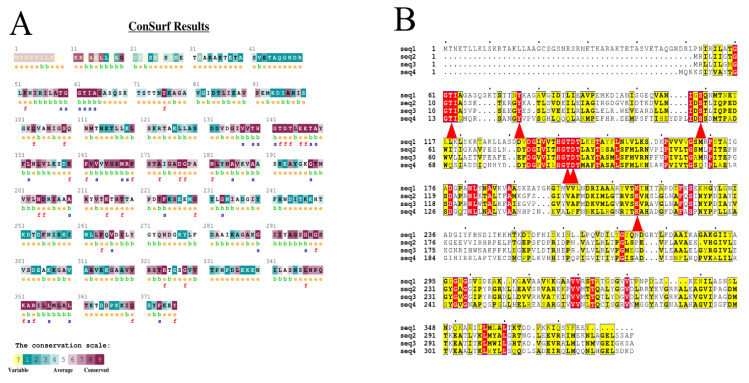
(**A**) Conservation analysis of BLASNase by ConSurf. Cyan-to-purple palette corresponds to unconservative (grade 1)-to-conserved (grade 9) scores. (**B**) Sequence alignment of BLASNase and other high enzyme activity ASNases. The active sites are marked by a red triangle below. Seq 1, BLASNase; seq 2, *Thermococcus gammatolerans* EJ3 ASNase (WP_015859055.1); seq 3, *Pyrococcus yayanosii* ASNase (WP_013906452.1); seq 4, *Pectobacterium carotovorum* MTCC 1428 ASNase (AFA36648.1).

**Figure 2 ijms-23-09663-f002:**
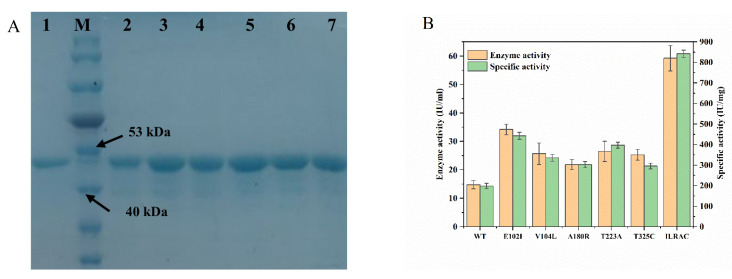
(**A**) SDS-PAGE analysis of BLASNase. M, protein molecular weight marker; lane 1, wild-type; lane 2, mutant E102I; lane 3, mutant V104; lane 4, mutant A180R; lane 5, mutant T223A; lane 6, mutant T325C; lane 7, mutant enzyme ILRAC (E102I/V104L/A180R/T223A/T325C). (**B**) Enzyme activity and specific activity of BLASNase.

**Figure 3 ijms-23-09663-f003:**
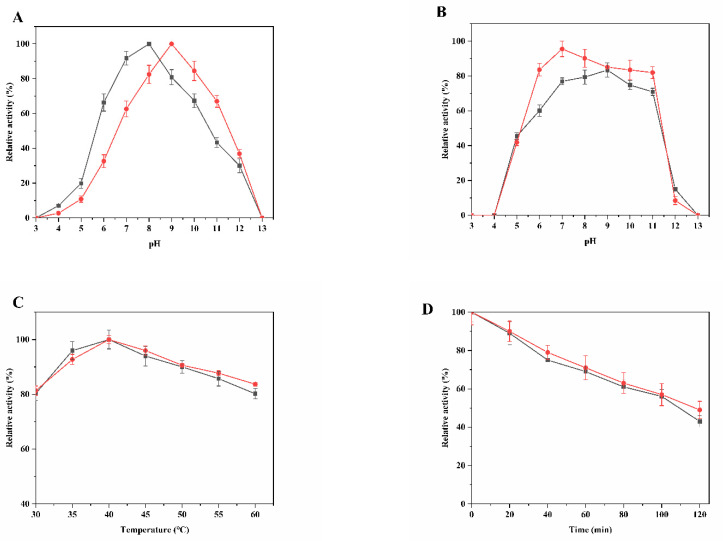
The temperature and pH effects on BLASNase: (**A**) relative activity at various pH; (**B**) residual activity after incubation at 4 °C for 24 h; (**C**) relative activity at various temperatures; (**D**) residual activity after incubation at 60 °C during 120 min. Black squares, wild-type; Redcircles, and mutant ILRAC.

**Figure 4 ijms-23-09663-f004:**
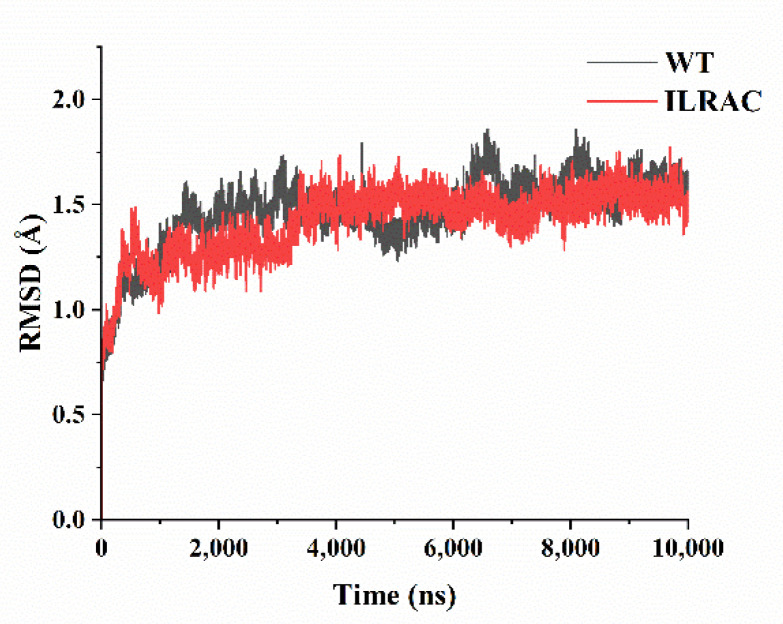
Conformational changes in the wild-type and mutant ILRAC during 100 ns MD (protein–ligand RMSD).

**Figure 5 ijms-23-09663-f005:**
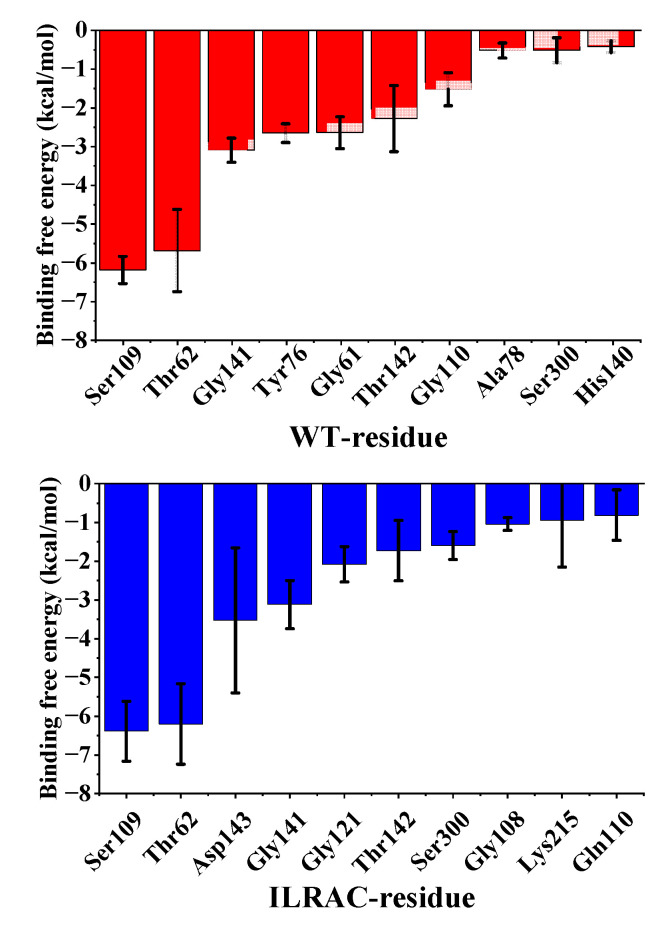
The top 10 residues that contribute to protein and substrate binding.

**Figure 6 ijms-23-09663-f006:**
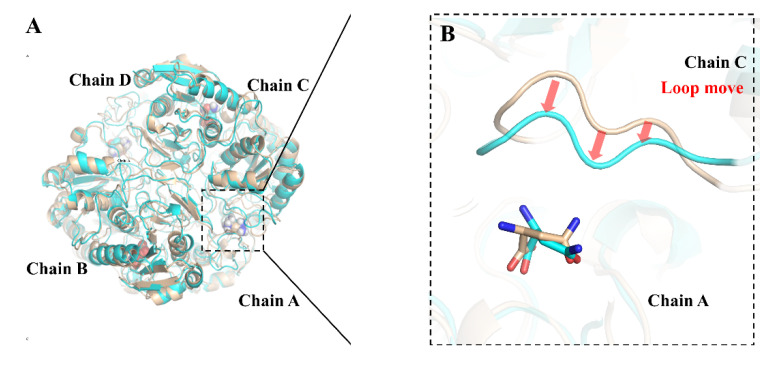
The binding patterns of wild-type and mutant ILRAC complexes with L-asparagine after 100 ns MD. The orange cartoon represents wild-type and the blue cartoon represents mutant ILRAC. (**A**) The overlay of wild-type and ILRAC. The sphere indicates the substrate L-asparagine. (**B**) Alterations in the substrate binding site. (**C**) The interaction of wild-type with L-asparagine. (**D**) The interaction of mutant ILRAC with L-asparagine. The yellow dashed line represents the hydrogen bond.

**Figure 7 ijms-23-09663-f007:**
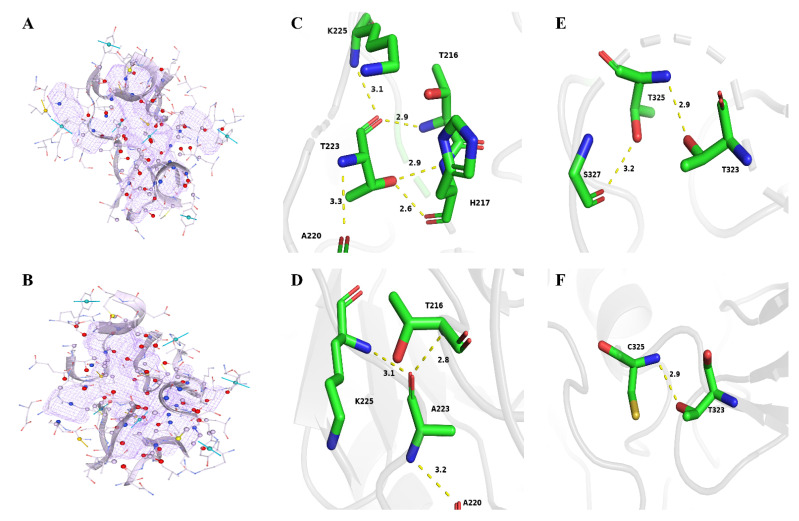
(**A**,**B**) Substrate binding pocket of wild-type and mutant ILRAC. (**C**,**D**) Change in the residue interactions around the 223rd residue. (**E**,**F**) Change in the residue interactions around the 325th residue.

**Figure 8 ijms-23-09663-f008:**
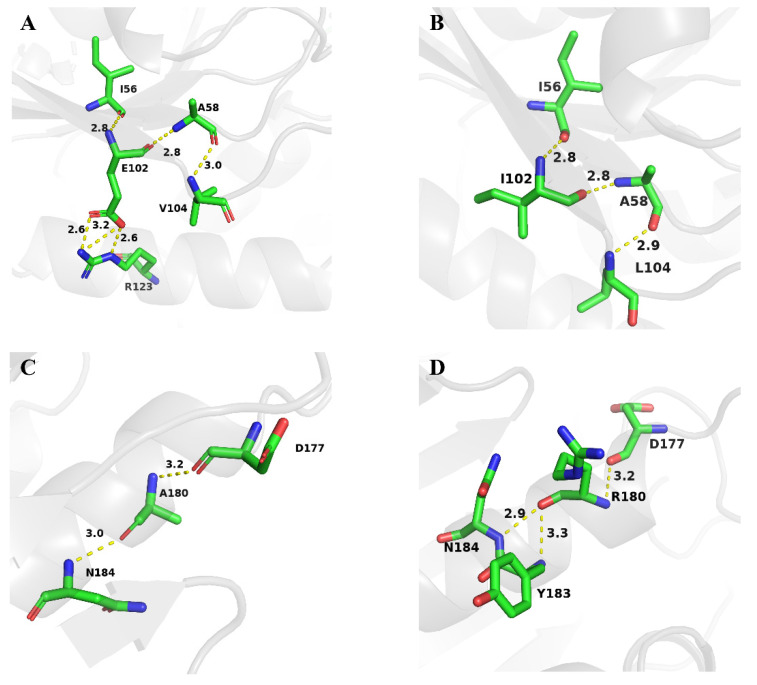
(**A**,**B**) Change in the residue interactions around the 102nd and 104th residue. (**C**,**D**) Change in the residue interactions around the 180th residue.

**Table 1 ijms-23-09663-t001:** Catalytic properties of wild-type and mutant ILRAC.

Enzymes	V_max_ (mmol·min^−1^·mg^−1^)	*K*_m_ (mM)	*k*_cat_ (min^−1^)	*k*_cat_/*K*_m_ (min^−1^·mM^−1^)
WT	9.93 ± 0.94	2.33 ± 0.49	197.95 ± 5.42	84.89
ILRAC	11.42 ± 1.63	1.45 ± 0.18	778.87 ± 36.43	537.15

**Table 2 ijms-23-09663-t002:** Secondary structures analysis of wild-type and mutant ILRAC by circular dichroism.

Secondary Structure	Proportion (%)	
	WT	ILRAC
α-Helices	26.0	34.5
β-Sheets	16.9	11.0
Turns	17.5	23.1
Random coli	39.6	31.4

**Table 3 ijms-23-09663-t003:** Hydrogen bonds number, Rg, and SASA values for wild-type and mutant ILRAC.

Enzymes	Hydrogen Bonds Number	Rg (Å)	SASA (Å^2^)
WT	810.63 ± 28.92	20.35 ± 0.08	36,276.87 ± 1323.39
ILRAC	812.60 ± 26.58	20.38 ± 0.07	36,675.61 ± 968.37

**Table 4 ijms-23-09663-t004:** Binding free energies and energy components calculated by MM/GBSA (kcal/mol).

System Name	WT-ASN	ILRAC-ASN
ΔE_vdw_ ^1^	−12.24 ± 3.34	−16.57 ± 2.85
ΔE_elec_ ^2^	−128.76 ± 5.85	−126.43 ± 7.69
ΔG_GB_ ^3^	111.02 ± 6.67	110.95 ± 6.18
ΔG_SA_ ^4^	−2.87 ± 0.09	−3.17 ± 0.09
ΔG_bind_ ^5^	−32.86 ± 2.48	−35.23 ± 2.52

^1^ ΔE_vdW_: van der Waals energy. ^2^ ΔE_elec_: electrostatic energy. ^3^ ΔG_GB_: electrostatic contribution to solvation. ^4^ ΔG_SA_: non-polar contribution to solvation. ^5^ ΔG_bind_: binding free energy.

**Table 5 ijms-23-09663-t005:** The size and shape description of substrate binding pockets.

	WT	ILRAC
volume (Å^3^)	1216.88	1248.40
surface (Å^2^)	1074.33	1100.43
depth (Å)	28.61	28.18

## Data Availability

The data presented in this study are available on request from the corresponding author.

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
