# Peer review of "Enhancing the Catalytic Activity of Type II L-Asparaginase from Bacillus licheniformis through Semi-Rational Design"

_ijms, 2022, doi:10.3390/ijms23179663_

Round 1

Reviewer 1 Report

In this manuscript, Zhou et al describe the engineering of type II ASNase to achieve higher enzymatic activity. ASNases have been used in treatment of ALL and lymphosarcoma as well as in degrading carcinogenic acrylamide in food processing. Authors undertook the effort to improve ASNase to to overcome low enzymatic stability and activity.   Overall there are strong points 1) Semi-rational approach to screen and develop ASNase with higher enzymatic activity 2) Extensive characterization of mutant and wild-type ASNase properties   However, there are concerns over the manuscript 1) While the authors show increased activity (six-fold higher Kcat/Km) and improved stability, its unclear how its better compared to alternative ASNases from E.coli etc or what is real world use could be. It would be useful to have a comparison of ASNase activity in a functional setting. An asaparginase activity assay with ALL patient samples or a food processing assay with acrylamide levels.   2) The authors should also address what insights the mutation profile could help in engineering other ASNases. Its unclear if these mutations are unique to BLASNase   Overall I think this paper provides a useful advance in engineering Type II ASNase. However, I think the authors need to support the functional relevance and application for the more active engineered BLASNase.

Reviewer 2 Report

This manuscript describes the semi-rational engineering of a type II L-asparaginase to improve the enzyme’s catalytic activity as well as the enzyme’s stability. The authors thoroughly describe the practical utility of this class of enzymes, the logic and procedures for selecting and performing the semi-rational mutagenesis, as well as the characterization of the mutant enzymes produced. The best-performing mutant produced in the study represents a significant improvement over the wild-type enzyme. I find this study to be valuable, thoroughly executed, and of interest to the field as a whole.

The authors do not provide sufficient experimental detail on how the enzyme activity was measured for any of the enzymes tested here. The asparaginase assay needs to be thoroughly described.

The English writing in this manuscript is strong, however there any several small writing/grammatical errors that appear throughout the manuscript. This manuscript needs to be thoroughly “proofread” and edited before it is published for a professional audience.

Throughout this manuscript, there needs to be a space between the number and “oC” when temperature values are reported.

Many genus and species names throughout this manuscript are not italicized, this needs to be fixed.

The authors use the term “super” to describe activities, stabilities, etc throughout this manuscript. This reads as somewhat unprofessional writing which the authors may want to rectify.

Introduction paragraph 2: It is unclear what the authors mean by “the complexity of catalysis by distal residues”, as by definition distal residues cannot participate in catalysis (although they clearly play a role in determining protein structure and therefore play a role in catalytic activity). This statement should be re-worded.

Results and discussion section 2.1, paragraph 1: The sentence “since sites closed to the active center and interacted with the substrate…” needs to be reworded.

Table 2 needs to be contained on a single page.

In table 2, “helixs” should be changed to “helices”

The authors state that the computationally predicted number of hydrogen bonds for their mutant vs the wild type (812 vs 810) indicated better stability and therefore supports experimental observations. I question whether this very small difference in what is a computationally predicted metric can really be ascribed to some improvement in stability. I have the same question about the SASA values, which seem to be very similar for the wild type and the improved mutant.

Materials and Methods, section 3.3: “transferred into BL21 (DE3) competent cells” should be changed to “transformed into BL21….”

Materials and Methods, section 3.4: No real information is provided here on how the combined mutants were generated. The authors should provide at least some experimental detail on how this was performed.
